# The Transposition of Insertion Sequences in Sigma-Factor- and LysR-Deficient Mutants of *Deinococcus geothermalis*

**DOI:** 10.3390/microorganisms12020328

**Published:** 2024-02-04

**Authors:** Ji Hyun Park, Sohee Lee, Eunjung Shin, Sama Abdi Nansa, Sung-Jae Lee

**Affiliations:** Department of Biology, Kyung Hee University, Seoul 02447, Republic of Korea; jhjh31@naver.com (J.H.P.); ahasohi@naver.com (S.L.); eunj@khu.ac.kr (E.S.); maniazmn91@gmail.com (S.A.N.)

**Keywords:** *Deinococcus geothermalis*, gamma irradiation, LysR family regulator, non-pigmented phenotype, oxidative stress by hydrogen peroxide, sigma factor, transposition of insertion sequences (ISs)

## Abstract

Some insertion sequence (IS) elements were actively transposed using oxidative stress conditions, including gamma irradiation and hydrogen peroxide treatment, in *Deinococcus geothermalis*, a radiation-resistant bacterium. *D. geothermalis* wild-type (WT), sigma factor gene-disrupted (∆*dgeo*_0606), and LysR gene-disrupted (∆*dgeo*_1692) mutants were examined for IS induction that resulted in non-pigmented colonies after gamma irradiation (5 kGy) exposure. The loss of pigmentation occurred because *dgeo*_0524, which encodes a phytoene desaturase in the carotenoid pathway, was disrupted by the transposition of IS elements. The types and loci of the IS elements were identified as IS*Dge2* and IS*Dge6* in the ∆*dgeo*_0606 mutant and IS*Dge5* and IS*Dge7* in the ∆*dgeo*_1692 mutant, but were not identified in the WT strain. Furthermore, 80 and 100 mM H_2_O_2_ treatments induced different transpositions of IS elements in ∆*dgeo*_0606 (IS*Dge5*, IS*Dge6*, and IS*Dge7*) and WT (IS*Dge6*). However, no IS transposition was observed in the ∆*dgeo*_1692 mutant. The complementary strain of the ∆*dgeo*_0606 mutation showed recovery effects in the viability assay; however, the growth-delayed curve did not return because the neighboring gene *dgeo*_0607 was overexpressed, probably acting as an anti-sigma factor. The expression levels of certain transposases, recognized as pivotal contributors to IS transposition, did not precisely correlate with active transposition in varying oxidation environments. Nevertheless, these findings suggest that specific IS elements integrated into *dgeo*_0524 in a target-gene-deficient and oxidation-source-dependent manner.

## 1. Introduction

Strains of the genus *Deinococcus* are mainly Gram-positive bacteria and comprise more than 100 species. *Deinococcus* species typically exhibit remarkable resistance to extreme stress conditions, including high radiation, dehydration, oxidation, starvation, and exposure to toxic substances [1,2]. The primary resistance mechanisms against cellular stress are related to the ability of *Deinococcus* strains to protect damaged deoxyribonucleic acid (DNA) and protein; these mechanisms include the cell wall structure and its components, genome organization and packaging, such as ring-like structures, active removal of toxic compounds, and DNA repair, and unique transcription factors for the regulation of gene expression [3,4]. A couple of *Deinococcus* strains are relatively well studied for their physiological protective mechanisms under oxidative stress. Various types of protective mechanisms have been described, including (1) pigment-dependent protection with carotenoids, (2) enzymatic defense mechanisms with catalase, peroxidase and superoxide dismutase, (3) metal-ion-dependent protection with the regulation of the iron/manganese ratio, (4) examples of well-known stress response regulators including the general antioxidative activator OxyR, superoxide response protein SoxRS, and the alternative sigma factor RpoS, and (5) the regulation of intracellular redox potential by thiol concentrations via cystine import and biosynthesis [2,3,4,5,6,7,8,9,10,11,12,13,14].

Bacterial genomes consist of several transposable elements (TEs), including insertion sequences (ISs). ISs are mobile genetic elements (MGEs) that move within different or individual DNA molecules or between other MGE vectors through copy-and-paste and cut-and-paste transposition [15]. Currently, more than 4500 bacterial ISs are identified as belonging to 29 families based on their distinct characteristics [16,17]. In general, bacterial IS elements are less than 3000 bp in length, consisting of transposase (Tpase) and terminal inverted repeats (TIRs) within both borders [17,18]. IS-specific Tpases are necessary for DNA cleavage and strand transfer to the target DNA, and TIR sequences are needed for Tpase binding. Direct repeat (DR) sequences are generated at the site of insertion [15,17,19]. Tpases are one of the main elements for classifying IS families. Many Tpases belong to aspartic acid (D) and glutamic acid (E) residue (DDE) motif-type IS elements. According to several reports, these IS elements are transposed to other locations by extreme stress, such as oxidative stress, high temperature, gamma irradiation, and DNA-damaging substances, resulting in the destruction of genes, effects on the promoter, and silencing in intergenic regions [20,21,22].

*Deinococcus geothermalis* is a thermophilic bacterium with an optimum growth temperature of 45–50 °C that forms orange-colored colonies [23]. Based on the ISfinder platform (https://isfinder.biotoul.fr/, accessed on 15 August 2023), the *D. geothermalis* genome contains 19 types of ISs in 73 IS elements [16,24]. In previous studies, the active transposition of ISs was detected using non-pigmented phenotypic selection on *D. geothermalis* wild-type (WT) strains, and several target gene-disrupted mutants such as Dps, LysR, and a cystine importer [25,26,27,28]. In a putative Dps protein-deficient mutant (∆*dgeo*_0257), oxidative stress induced the integration of IS*Dge7* of the IS*5* family, resulting in the interruption of phytoene desaturase (Dgeo_0524, *crtI*), which is a carotenoid enzyme and protective factor against oxidative stress; the reddish-colored *D. geothermalis* transformed into a non-pigmented strain [25]. A LysR-type transcriptional regulator (LTTR) family *dgeo*_2840-gene-disrupted mutant strain (∆*dgeo*_2840) was selected from the non-pigmented colonies, and the transpositional integration of IS*Dge6* of the IS*5* family was found to have occurred [26]. ∆*dgeo*_1985R, a cystine-importer over-expressed mutant, and the cystine-importer-disrupted mutant ∆*dgeo*_1986-87, exhibited different IS transpositions of IS*Dge2* (IS*1* family)/IS*Dge5* (IS*701* family) and IS*Dge5*/IS*Dge7* (IS*5* family) in oxidative stress conditions, respectively [27]. Therefore, redox-imbalanced conditions and oxidative stress influenced the active transposition of ISs in *D. geothermalis*.

Sigma factor is a component of the multi-subunits of holoenzyme in bacterial ribonucleic acid (RNA) polymerase, which regulates the expression of genes through the redirection of transcription initiation [28]. An appropriate regulation of gene expression is required to survive under extreme conditions; this regulation is controlled by the associations between alternative sigma factors and RNA polymerase. Because sigma factor regulons consist of several genes, they contribute to the regulation of various bacterial genes as global regulators [29,30]. In general, the stress-responsive sigma factors σ^s^ and σ^B^ activate the expression of genes required for bacterial survival and cell viability under extreme stress conditions [30,31,32,33,34,35].

Currently, the functional roles of the putative sigma factor Dgeo_0606 of the extracytoplasmic function (ECF) family and the presumed LysR family regulator Dgeo_1692 are uncharacterized in *D. geothermalis*. Thus, we constructed *D. geothermalis* target-gene-deficient mutants Δ*dgeo*_0606 and Δ*dgeo*_1692 as *dgeo*_0606- and *dgeo*_1692-gene-disrupted mutants, respectively, determined physiological changes and detected non-pigmented phenotype selection for IS transposition under oxidative stress conditions, and identified the types of IS elements and their action modes. In this study, our aim was to elucidate the active transposition of IS elements by examining the physiological roles of *dgeo*_0606 and *dgeo*_1692 under oxidative stress conditions (specifically, gamma irradiation and H_2_O_2_ treatment).

## 2. Materials and Methods

### 2.1. Bacterial Strains and Culture Conditions

The wild-type (WT) strain *D. geothermalis* DSM11300^T^ (KACC12208) was obtained from the Korean Agricultural Culture Collection (KACC, National Academy of Agricultural Science, Republic of Korea; http://genebank.rda.go.kr, accessed on 1 December 2016). *D. geothermalis* is commonly cultured on tryptone glucose yeast extract (TGY) medium consisting of 10 g tryptone, 1 g glucose, and 5 g yeast extract per liter of distilled water at 48 °C. *Escherichia coli* DH5α was used for the transformation of recombinant plasmids and grown on Luria–Bertani (LB) medium consisting of 10 g tryptone, 5 g yeast extract, and 10 g NaCl per liter of distilled water at 37 °C (MB Cell, Kisan Bio Co., Seoul, Republic of Korea).

### 2.2. Construction of the ∆dgeo_0606 and ∆dgeo_1692 Mutant Strains

The target-gene-disrupted mutant strains ∆*dgeo*_0606 and ∆*dgeo*_1692 were con structed by the homologous recombination of a kanamycin resistance cassette encompassing identical sequence regions to the target gene; integration into the target gene was carried out according to our previous study [25]. For the ∆*dgeo*_0606 mutant strain, homologous DNA sequences of roughly 1.0 kb in length, from both sides of *dgeo*_0606, were amplified from genomic DNA and purified using an AccuPrep^®^ PCR purification kit (Bioneer Corp., Daejeon, Republic of Korea). First, the purified right-border DNA fragments and plasmid pKatAPH3 were cleaved by *Xba*I-*Pst*I and ligated into a plasmid (pKR0606). Then, to yield pKRL0606 as a left-border DNA fragment ligation, the purified left-border DNA fragments and plasmid pKR0606 were digested with *Kpn*I-*Sal*I, ligated, and propagated in *E. coli* DH5α. The final recombinant plasmid pKRL0606 was purified from *E. coli* and transformed into WT *D. geothermalis*-competent cells using a CaCl_2_-dependent technique described previously by Kim et al. [12]. The resulting transformed strain was named ∆*dgeo*_0606 mutant and selected on TGY agar containing 8 µL/mL kanamycin after incubation at 48 °C for 2 days. The transformed clone was confirmed through PCR detection of the target gene, resulting in extended length comparable to that of an antibiotic marker. The complementary strain was obtained using transformation of the expression shuttle vector pRADgro derivate containing a full-length gene of *dgeo*_0606 [36]. For the additional analysis involving comparison with the complementary strain of ∆*dgeo*_0606 (∆*dgeo*_0606C), the empty vector pRADgro was transformed into WT and ∆*dgeo*_0606 mutant strains of *D. geothermalis*. The resulting strains were indicated as WT/pRADgro and ∆*dgeo*_0606/pRADgro. The complementary strain of ∆*dgeo*_1692 mutant was constructed according to the same procedure, and the resulting transformed clone was detected by PCR analysis. However, unfortunately, the complemented strain for ∆*dgeo*_1692 mutant was not constructed, and the empty vector pRADgro was not transformed.

### 2.3. Growth Curves of Wild-Type and Mutant Strains

To evaluate the growth curves of the WT, ∆*dgeo*_0606, ∆*dgeo*_1692, and ∆*dgeo*_0606C strains, each strain was grown overnight in TGY broth at 48 °C. Then, the strains were inoculated to an optical density at 600 nm (OD_600_) of 0.06 and incubated at 48 °C with shaking (180 rpm). The continuous growth of the strains was monitored hourly using the 1/10 dilution measurement method from previous work [36]. In general, the maximum OD_600_ was saturated at 8–10 levels (OD_600_ = 8.0 as stationary phase), and two different growth phases of OD_600_ = 2.0 and 4.0 were chosen as the early-exponential and late-exponential phases, respectively.

### 2.4. Viability Test in H_2_O_2_ Oxidative Stress Conditions

The WT, ∆*dgeo*_0606, and ∆*dgeo*_1692 mutant strains of *D. geothermalis* were grown to an OD_600_ of 2.0 or 4.0 in TGY broth at 48 °C. To ensure an equivalent number of cells from each culture, cell densities were adjusted to an OD_600_ of 1.0, treated with 80 and 100 mM H_2_O_2_, and continuously cultured for 1 h at 48 °C. After 1 h, the samples were serially diluted 10-fold in buffered saline from 10^−1^ to 10^−5^. A 5 µL volume of each diluted suspension was spotted and incubated on the TGY agar plates at 48 °C.

### 2.5. Gamma Radiation Treatment Conditions

Two growth-phase point cells at an OD_600_ of 2.0 and 4.0 were prepared for exposure to gamma irradiation. Bacterial samples were harvested, resuspended with 0.9% NaCl solution, and normalized to OD_600_ 1.0. Each 3 mL sample was poured into a 5 mL microcentrifuge tube and exposed to a total radiation dose of 5 kGy from a cobalt-60 (^60^Co) radiation generator (Advanced Radiation Technology Institute of Korea Atomic Energy Research Institute (KAERI), Jeongup, Republic of Korea). The exposed samples were then diluted from 10^−1^ to 10^−5^, and 100 µL was plated on TGY agar. The colony-forming units (CFUs) were enumerated following a 2-day incubation period at 48 °C.

### 2.6. Selection of Non-Pigmented Colonies and Detection of Transposition of Insertion Sequences

After treatment with 5 kGy gamma irradiation or 80 and 100 mM H_2_O_2_ for 1 h, the samples were diluted, plated on TGY agar, and incubated at 48 °C for 2 days. Non-pigmented colonies were isolated and re-cultured in TGY broth. Genomic DNA was extracted from the 5 mL cultured non-pigmented cells using the HiYield™ Genomic DNA Mini Kit (Real Biotech Corp., Taipei County, Taiwan). PCR was performed using ExTaq^®^ DNA polymerase (TaKaRa Bio Inc., Shiga, Japan) coupled with two primers encompassing carotenoid biosynthesis genes, including *dgeo*_0523 for phytoene synthase and *dgeo*_0524 for phytoene desaturase, to detect the novel IS locus that occurred by IS transposition. The PCR products were separated by electrophoresis in a 1% agarose gel, and the enlarged PCR products were purified using the AccuPrep^®^ PCR purification kit. DNA sequencing of the PCR products was performed by 454 DNA sequencing (Macrogen Co., Seoul, Republic of Korea).

### 2.7. Analysis of IS Types in Transposition Events

Each IS element was detected and determined by DNA sequencing analysis. IS typing was analyzed using the bacterial IS detection platform ISfinder [16] and laboratory data [24]. DNA sequences were aligned and their identities were calculated via BLAST of NCBI. The conserved DNA sequences of TIR and DR were detected and compared to those in the ISfinder platform to determine the IS type. To determine the transposition action mode, identical IS loci genes were amplified by primer sets of unique target genes beyond the boundaries of full-length IS elements.

### 2.8. Quantitative RT-PCR Analysis for Measurement of Gene Expression Level

To determine the gene expression levels of antioxidant-related proteins, such as catalase (*dgeo*_2728) and cystine-importer components (*dgeo*_1986 and *dgeo*_1987), quantitative RT-PCR (qRT-PCR) was performed on the WT, ∆*dgeo*_0606, and ∆*dgeo*_1692 mutant strains. The expression levels of induced transposases were also compared between WT and ∆*dgeo*_0606 because IS transpositions were exhibited in both strains as a result of hydrogen peroxide stress. Cells were incubated to an OD_600_ of 2.0 with present or absent 50 mM H_2_O_2_, which resulted in oxidative stress without cell death, and total RNA was extracted using an RNA extraction kit (Qiagen GmbH, Hilden, Germany). cDNA was prepared by cDNA synthetase using Prime^TM^ reverse transcriptase (TaKaRa Bio Inc., Shiga, Japan). qRT-PCR was performed by the CFX Connect Real-Time PCR Detection System (BioRad Laboratories, Inc., Hercules, CA, USA). Data evaluation was conducted by analysis of variance (ANOVA) *t*-tests, using data from three replicates.

## 3. Results

### 3.1. Construction of the ∆dgeo_0606 and ∆dgeo_1692 Mutants of D. geothermalis

Two target-gene-disrupted mutant strains—∆*dgeo*_0606, deficient of an alternative sigma factor, and ∆*dgeo*_1692, deficient of a putative LysR family regulator—were constructed using a homologous recombination procedure and confirmed using PCR analysis (Figure 1). These constructs were generated and subsequently replaced through integration with the kanamycin-resistant *aph* gene via homologous recombination involving approximately 1 kb segments from both border regions of the *dgeo*_0606 and *dgeo*_1692 loci (Appendix A). The mutant clones with kanamycin resistance were selected, and displayed PCR products larger than those of the WT parent strain (Figure 1).

The growth patterns were compared between the WT and the mutant strains (∆*dgeo*_0606 and ∆*dgeo*_1692) in TGY medium at 48 °C. Figure 2A shows growth curves based on three independent replicate experiments. At first, there was a slight delay in the growth of the ∆*dgeo*_1692 strain compared to the WT strain for about an hour; however, it exhibited a similar growth pattern to the WT. In contrast, ∆*dgeo*_0606 displayed a much slower growth curve than the WT for the first 5 h, suggesting that the deletion of *dgeo*_0606 caused several physiological defects. However, after that, the growth curve of ∆*dgeo*_0606 increased exponentially and overtook the maximal growth level of ∆*dgeo*_1692 after 11 h. The ∆*dgeo*_0606C strain did not recover from growth delay during the early growth phase. Based on qRT-PCR assays for *dgeo*_0606 and *dgeo*_0607 at different growth phases, the ∆*dgeo*_0606 strain exhibited an overexpression of *dgeo*_0607, possibly acting as an anti-sigma factor (Appendix A).

For physiological analysis based on the growth phase, we chose two time points: the early-exponential (E) phase with an OD_600_ of 2.0, and the late-exponential (L) phase with an OD_600_ of 4.0. For the effect of oxidative stress on the WT and mutants, E-phase cells were treated with H_2_O_2_ at concentrations of 80 and 100 mM for 1 h before a viability test was conducted (Figure 2B). The WT and ∆*dgeo*_1692 showed similar viability. However, the viability of ∆*dgeo*_0606 was more resistant than that of the WT under 80 and 100 mM H_2_O_2_ treatment at the E phase (upper panel). The WT strain exhibited less susceptibility under 80 and 100 mM H_2_O_2_ treatment compared to E phase. In contrast, ∆*dgeo*_0606 lost viability at the L phase, while the complementary strain regained viability with the expression vector pRADgro base (lower panel). The pRADgro vector did not directly affect viability against H_2_O_2_ treatment. This resulted in viability levels exhibiting some flexibility regarding the inhibitory concentration of H_2_O_2_ in independent trials. Nonetheless, in summary, the growth and viability patterns depended on the expression of *dgeo*_0606 and its antagonist, possibly a redox potential substance or anti-sigma factor, in a growth-phase-dependent manner. The expression levels of a cystine ABC transporter (*dgeo*_1986-87), acting as a redox imbalance protector, and catalase (*dgeo*_2728), functioning as a direct blocker of H_2_O_2_, were examined and are explained later in the text.

### 3.2. Selection of Non-Pigmented Clones by Gamma-Irradiation from WT and Mutant Strains

*Deinococcus* strains generally exhibit a phenotypic reddish color resulting from carotenoids, the pigments involved in their antioxidant stress response. According to the KEGG pathway database (Kyoto Encyclopedia of Genes and Genomes, https://www.genome.jp/kegg/pathway.html, accessed on 1 July 2022), the non-pigmented phenotype is likely due to the dysfunction of a key enzyme in the carotenoid biosynthesis pathway (Appendix A) [37]. The stress effect of gamma irradiation (5 kGy) on the cell growth of the WT and mutants (∆*dgeo* 0606 and ∆*dgeo*_1692) was confirmed by diluting the exposed cultures to 10^−4^ and plating them on TGY agar. The survival ability of each was represented using CFU counts. All three samples, WT, ∆*dgeo*_0606, and ∆*dgeo*_1692 strains, exhibited greater resistance in the E phase than in the L phase. However, both mutant strains showed better viability than the WT after gamma irradiation stress; especially, ∆*dgeo*_0606 exhibited 5-fold more resistance than ∆*dgeo*_1692. Non-pigmented phenotypic colonies may have been detected because the carotenoid biosynthesis pathway was disrupted through IS transposition and other mutations. There were nine Δ*dgeo*_0606 non-pigmented colonies, five at the E and four at the L phases, and two colonies from the Δ*dgeo*_1692 mutant at the L phase (Figure 3; Appendix A). However, the WT strain did not exhibit non-pigmented mutation from gamma irradiation at 5 kGy in the present work.

Based on the CFU data, the yield frequency of non-pigmented colonies was calculated for Δ*dgeo*_0606 and Δ*dgeo*_1692 mutants. The mean frequencies of non-pigmented colonies were 2.31 × 10^−6^ for Δ*dgeo*_0606 at the E phase, 4.93 × 10^−5^ for Δ*dgeo*_0606 at the L phase, and 9.1 × 10^−5^ for Δ*dgeo*_1692 at the L phase. There was no non-pigmented colony for Δ*dgeo*_1692 at the E phase.

### 3.3. Detection of IS Transposition Events in Non-Pigmented Strains

Two genes, *dgeo*_*0523* and *dgeo*_*0524,* were identified as possible transposition target genes, as they are involved in the carotenoid pathway and because the other five genes were not changed in the non-pigmented strains (Appendix A). PCR analysis was performed for each target gene using a comprehensive primer set to assess the active transposition of IS elements. The amplified PCR products of the carotenoid biosynthesis genes were observed to enlarge in size with the integration of IS elements when compared with PCR products with WT as a negative control. The results indicated that there was no integration in *dgeo*_0523. However, the *dgeo*_0524 gene, which encoded phytoene desaturase, was found to be dysfunctional due to the integration of IS elements in eight isolates (Figure 3A). Interestingly, three non-pigmented strains did not have any mutations in phytoene desaturase. This result implies that the non-pigmented phenotype of *D. geothermalis* arises from gene disruption caused by IS element integration into phytoene desaturase within the carotenoid synthesis pathway, triggered by the gamma-irradiation-induced active transposition of several IS elements.

### 3.4. Determination of IS Type and Transposition Loci under Gamma-Irradiation

Based on the electrophoresis results of the PCR products, there were at least two different IS integrations on the *dgeo*_0524 gene in the non-pigmented mutant strains (Figure 3A). To determine the IS type and transposition loci on the non-pigmented Δ*dgeo*_0606*gw* and Δ*dgeo*_1692*gw* mutant strains, the DNA sequences were analyzed. Appendix A shows the insertion sites and classified IS elements from the integrated DNA sequence analysis.

In the Δ*dgeo*_0606*gw* strain, non-pigmentation occurred with the integration of two types of IS elements. In four non-pigmented strains, integrations of IS*Dge6* of the IS*5* family were found at *dgeo*_0524’s 8th and 319th positions; in the other two non-pigmented strains, the integration of IS*Dge2* of the IS*1* family was found at the 1289th position. The comparative locations of the IS elements’ integration in Δ*dgeo*_0606*gw* are shown in Figure 3B.

In the Δ*dgeo*_1692*gw* strain, there were two types of IS element integrations on each of the non-pigmented strains. The first was the integration of IS*Dge7* of the IS*5* family at the 1673rd position of *dgeo*_0524. The other was the integration of IS*Dge5* of the IS*701* family at the 612th position (Figure 3B).

### 3.5. The Effect of H_2_O_2_ Treatment on WT, ∆dgeo_0606, and ∆dgeo_1692

Similar to determining the effects of gamma irradiation, the effect of oxidative stress as a result of H_2_O_2_ exposure was determined for the WT, ∆*dgeo*_0606, and ∆*dgeo*_1692 strains. Based on the results of the viability tests, non-pigmented colony selection, and IS detection in each strain, the effect of H_2_O_2_ oxidative stress was compared to gamma irradiation treatment. The survival ability of each strain was determined using CFU counts. Based on the CFU data, the frequency of non-pigmented colonies was calculated for each strain as follows: WT, 5.6 × 10^−5^ at 80 mM H_2_O_2_; Δ*dgeo*_0606 mutant, 3.12 × 10^−6^ at 80 mM and 3.1 × 10^−5^ at 100 mM H_2_O_2_; and Δ*dgeo*_1692, 7.96 × 10^−6^ at 80 mM H_2_O_2_. Therefore, the frequency of non-pigmented production as a result of H_2_O_2_ treatment was approximately 10-fold less than that of gamma irradiation.

The H_2_O_2_-treated strains (80 and 100 mM) at both OD_600_ 2.0 and 4.0 were diluted to 10^−4^ and 10^−5^ and plated on TGY medium. After 2 days of incubation, non-pigmented phenotypic colonies were isolated from WT, ∆*dgeo*_0606, and ∆*dgeo*_1692 as follows: two colonies from the WT, three from the Δ*dgeo*_0606 mutant, and one from the Δ*dgeo*_1692 mutant. All of these non-pigmented colonies came from H_2_O_2_-treated strains at OD_600_ 2.0. The Δ*dgeo*_0606 mutant’s non-pigmented isolates exhibited a disruption to the *dgeo*_0524 gene with the integration of IS elements; however, Δ*dgeo*_1692*hw* did not (Figure 4A and Appendix A). IS*Dge6* of the IS*5* family was integrated at the eighth position of *dgeo*_0524 in the WT strain (Figure 4A).

In the ∆*dgeo*_0606*hw* mutant, IS*Dge6* and IS*Dge7* of the IS*5* family and IS*Dge5* of the IS*701* family were integrated at the 723rd, 1403rd and 1450th positions on the *dgeo*_0524 gene, respectively (Figure 4B and Appendix A). For the non-pigmented colony, Δ*dgeo*_1692*hw* from Δ*dgeo*_1692, other carotenoid pathways involving genes were tested for the detection of IS integration. However, these genes, including *dgeo*_0857, *dgeo*_1618, *dgeo*_2306, and *dgeo*_2309, as well as dgeo_2310 as a FAD-dependent oxidoreductase, did not exhibit IS transposition (Appendix A). Therefore, the results suggest that the Δ*dgeo*_1692*hw* mutant may have other dysfunctions in carotenoid biosynthesis, including point mutations.

### 3.6. Gene Expression Levels of Catalase, Cystine Importers, and Transposases

To characterize the expression levels of cystine-binding protein, ABC transporter permease for cystine uptake, and catalase as a redox protector and an H_2_O_2_ cleaner under H_2_O_2_ conditions (50 mM as an oxidative stress condition without cell death), qRT-PCR was conducted for each gene of the WT, ∆*dgeo*_0606, and ∆*dgeo*_1692 strains. The expression levels of cystine-binding protein and permease were up-regulated at an OD_600_ of 2.0, but mostly down-regulated at an OD_600_ of 4.0 after H_2_O_2_ treatment. Interestingly, cystine-binding protein and permease were strongly up-regulated in the WT and ∆*dgeo*_0606 strains at an OD_600_ of 2.0 under H_2_O_2_ conditions. Catalase was up-regulated in WT at OD_600_ but down-regulated in ∆*dgeo*_0606 at an OD_600_ of 4.0 (Figure 5A).

In addition, to assess the expression levels of transposases within IS*Dge2*, IS*Dge5*, IS*Dge6*, and IS*Dge7* elements under conditions of 50 mM H_2_O_2_, qRT-PCR analyses were performed for each transposase family in both the WT and ∆*dgeo*_0606 mutant strains. This was undertaken due to the observed active transposition of IS elements induced by H_2_O_2_ treatment in both strains. The expression levels of IS*Dge2*, IS*Dge5*, IS*Dge6*, and IS*Dge7* were all up-regulated at an OD_600_ of 2.0. However, they showed different expression level patterns at an OD_600_ of 4.0; notably, IS*Dge7* was down-regulated in the WT and ∆*dgeo*_0606 mutant strains (Figure 5B).

## 4. Discussion

The transposition of IS elements can occur as a result of intrinsic cellular regulations and extracellular stress conditions, such as UV, radiation, oxidation, and heat shock. Active IS transposition has been known to introduce unspecific events, including the type of IS family transposed into the genome loci and severe gene destruction. Exceptionally, the IS*5* family integrated into the unique site of the stress-induced DNA destabilization (SIDD) region near the promoter of the *fucPIK* and *glpF* genes in *Escherichia coli* [38,39,40]. Nevertheless, the transposition mechanisms of several IS elements have been well established using different Tpase motifs; for example, DDE, histidine-hydrophobic-histidine (HuH) motifs, and the regulatory system of the IS*1* family comprising Tpase A and Tpase B have been highlighted [18,41,42].

In a previous study, we selected non-pigmented colonies of WT *D. geothermalis*, a DNA-binding protein (Dps) gene-disrupted mutant (∆*dgeo*_0257), and a LysR gene-disrupted mutant (∆*dgeo*_02840) under oxidative stress conditions. Dgeo_0524, the main enzyme of the carotenoid pathway, was disrupted by the active transposition of IS elements in the non-pigmented colonies (IS*Dge11* in the wild-type, IS*Dge6* in the ∆*dgeo*_2840 mutant, and IS*Dge7* in the ∆*dgeo*_0257 mutant) [25,26]. In addition, non-pigmented colonies were selected from the WT, Dps-deficient (∆*dgeo*_0257 and ∆*dgeo*_0281) mutants, an OxyR-deficient (∆*dgeo*_1888) mutant, a putative TrmB-deficient (∆*dgeo*_1985R) mutant, and a cystine-importer-deficient (∆*dgeo*_1986-87) mutant under stress conditions induced by gamma irradiation [28]. Both the *dgeo*_0524 and *dgeo*_0523 genes of the carotenoid pathway were disrupted by the transposition of several types of IS elements, namely IS*Dge6* in the WT, ∆*dgeo*_1888, ∆*dgeo*_1985R, ∆*dgeo*_0257, and IS*Dge2* in ∆*dgeo*_0281, and IS*Dge5* and IS*Dge6* in ∆*dgeo*_1986-87.

In this study, we observed the active transposition of IS elements after gamma irradiation and H_2_O_2_ treatments in a sigma factor gene-disrupted mutant (∆*dgeo*_0606) and a putative *LysR* gene-disrupted mutant (∆*dgeo*_1692). After strains were exposed to gamma irradiation of 5 kGy, eleven non-pigmented colonies were selected: nine colonies from the ∆*dgeo*_0606 mutant and two from ∆*dgeo*_1692. In eight non-pigmented colonies, the transposition of IS elements occurred in *dgeo*_0524. In the ∆*dgeo*_0606*gw* strain, IS*Dge6* and IS*Dge2* were incorporated into the *dgeo*_0524 gene by transposition. In the ∆*dgeo*_1692*gw* strain, IS*Dge5* and IS*Dge7* were incorporated into *dgeo*_0524 (Figure 3B). This suggests that *dgeo*_0524 might be a target gene for the integration of several IS elements, resulting in phenotypical non-pigmented colonies.

Furthermore, we observed IS transposition under H_2_O_2_ oxidative stress in the same mutant strains (∆*dgeo*_0606 and ∆*dgeo*_1692), with six non-pigmented colonies in total: two for WT, three for ∆*dgeo*_*0606*, and one for ∆*dgeo*_1692. Among them, ∆*dgeo*_1692*hw* did not exhibit IS transposition in *dgeo*_*0524*, suggesting that loss of pigmentation may be the result of a point mutation in carotenoid synthesis genes, except for *dgeo*_0524. Figure 4 shows IS transposition in WT*hw* and ∆*dgeo*_0606*hw*. Interestingly, ∆*dgeo*_0606*hw2* revealed IS*Dge7* transposition under 100 mM H_2_O_2_ treatment because IS*Dge7* was transposed on the ∆*dgeo*_1692 mutant by gamma irradiation. The active transposition of IS element families was dependent on target gene disruptions and growth phases because a putative sigma factor, *dgeo*_0606, and a cystine-uptake transporter, *dgeo*_1986-87, exhibited significantly different expression levels in the E- and L-exponential growth phases, respectively (Figure 5) [27]. Therefore, IS selectivity due to oxidative stress and the determination of the integration loci further resolve challenges in studying IS transposition in *D. geothermalis*.

Based this study’s data, IS elements inserted in transposition loci are influenced by several conditions, including extracellular stress and the type of disrupted gene [43]. Therefore, the question is whether the specific IS elements involved in transposition depend on intrinsic factors, such as a sigma factor and LysR, and extracellular stress conditions, such as oxidative H_2_O_2_ and gamma irradiation stress. The LysR-type transcriptional regulator (LTTR) family are transcriptional regulators that regulate the expression of various genes and regulons in many prokaryotes [44,45]. Furthermore, a sigma factor regulates the appropriate gene expression and protein synthesis under extreme stress conditions [46]. However, both LysR and a sigma factor can regulate various genes. Therefore, further research about the role of LysR (∆*dgeo*_1692) and a sigma factor (∆*dgeo*_0606) under oxidative stress and gamma irradiation is needed. Using previous RNA sequencing (RNA-seq) and qRT-PCR analysis results for ∆*dgeo*_1986-87, the expression levels of catalase, cystine importers, and Tpases in various IS elements were identified for the network regulation of the ∆*dgeo*_0606 and ∆*dgeo*_1692 mutants (Figure 5) [27,36]. In addition, there could be a meaningful correlation between the LysR family, sigma factors, and the active transposition of specific IS types. Thus, further research is required to identify the biological roles of the LysR family and sigma factors in *D. geothermalis* under stress conditions, especially regarding the functional regulatory mechanism between a putative sigma factor, *dgeo*_0606, and a possible anti-sigma factor, *dgeo*_0607, which may impede the recovery of delayed growth [47]. To mitigate the side effects of inducing the neighbor gene, we need counter-transcription directional gene disruption because the promoter of the antibiotic-resistant gene affects transcription levels, similar to ∆*dgeo*_1985R construction [13].

Using the bacterial promoter prediction tool BacPP (http://www.bacpp.bioinfoucs.com/home) to forecast controlling sigma factors [48], it was determined that Tpase promoters of IS*Dge3* and IS*Dge11* are targets of sigma factor σ^28^, exhibiting a recognition rate of over 60% significant, while IS*Dge2* is targeted by σ^24^ with a significant recognition rate of 56% in WT *D. geothermalis*. Conversely, Tpase promoters of IS*Dge5*, IS*Dge6*, and IS*Dge7* are predicted to be regulated by σ^24^ with a recognition rate of 10–15%. However, despite these prediction, IS*Dge2*, IS*Dge5*, IS*Dge6*, and IS*Dge7* elements were observed to undergo active transposition in the ∆*dgeo*_0606 strain under gamma irradiation and H_2_O_2_ treatment in this study. Interestingly, the expression levels of these three IS transposases were induced specifically during the E growth phase in both WT and ∆*dgeo*_0606 mutant strains upon H_2_O_2_ treatment. This suggests that *dgeo*_0606 may not function as the exact sigma factor σ^24^ for Tpase induction. Actually, we still do not know what the inducing signaling factor is for Tpase, as a key player in active transposition. Therefore, understanding the signaling procedures and network regulations for active IS transposition among Tpases, regulators, and signal resources is an important aspect of bacterial genomic plasticity and evolution.

## 5. Conclusions

The radiation-resistant bacterium *D. geothermalis* exhibited the active transposition of IS elements under various oxidative stress conditions, such as gamma irradiation and hydrogen peroxide treatment, in sigma-factor- and LysR-transcription-factor-deficient mutants. Differential IS family members induced and triggered active transposition in the replicative mode of action. The transposition frequency of hydrogen peroxide treatment was 10-fold less than that of gamma irradiation. Nevertheless, basically, the probability of isolation of non-pigmented colonies is limited to dysfunction in the carotenoid biosynthesis pathway, which is the phenotypic selection in the present study. There are many open questions about the IS-type selectivity of various environmental resources, the gene-gain and -loss concepts of genomic plasticity, the signaling networks between signal reception and the induction of major enzyme transposases in IS transposition, and the detailed transposition mechanisms of individual IS elements. This field of research, exploring the presence and behavioral properties of IS elements found in bacterial genomes in the last half century, is therefore a fairly new area in the study of microbial genetics.

## Figures and Tables

**Figure 1 microorganisms-12-00328-f001:**
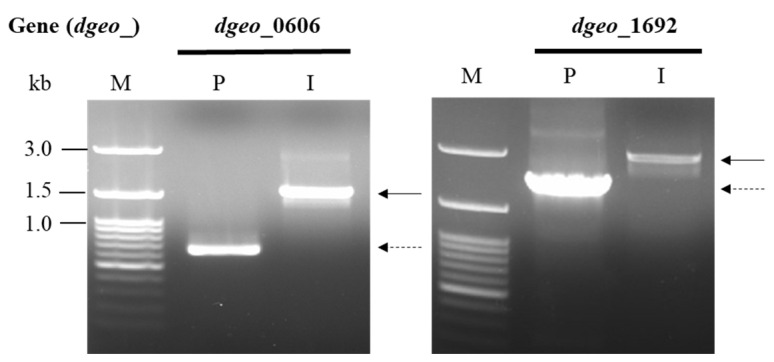
PCR detection by primer sets for *dgeo*_0606 and *dgeo*_1692. Lanes: M, size marker; P, parent (WT); I, insertion of kanamycin-resistant gene (*aph*) in both *dgeo*_0606 and *dgeo*_1692 genes. Dash line is an intact gene length and solid line is an extended PCR product by Km-resistant gene insertion. The size of *aph* gene product is ca. 850 bp.

**Figure 2 microorganisms-12-00328-f002:**
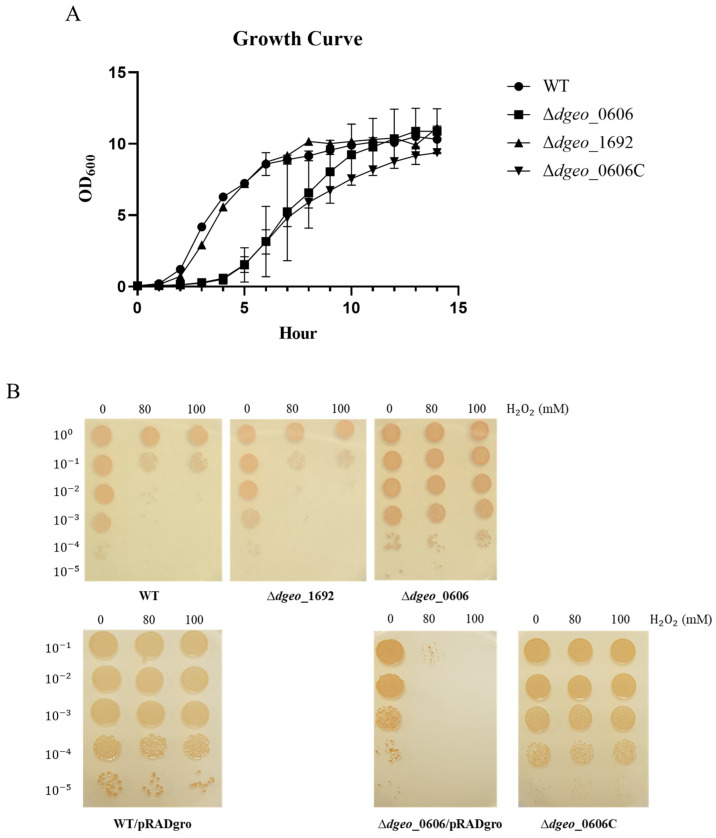
(**A**) Growth curves of the WT, ∆*dgeo*_0606, ∆*dgeo*_1692, and ∆*dgeo*_0606C strains in TGY medium. The growth curves are represented as the average of three independent replicates. (**B**) Viability test of the WT and two mutant strains, ∆*dgeo*_0606 and ∆*dgeo*_1692, treated with hydrogen peroxide of 0, 80, and 100 mM at early-exponential phase, OD_600_ = 2.0 (**upper panel**), compared with late-exponential growth phase, OD_600_ = 4.0, with expression vector pRADgro for complementary strain ∆*dgeo*_0606C (**bottom panel**).

**Figure 3 microorganisms-12-00328-f003:**
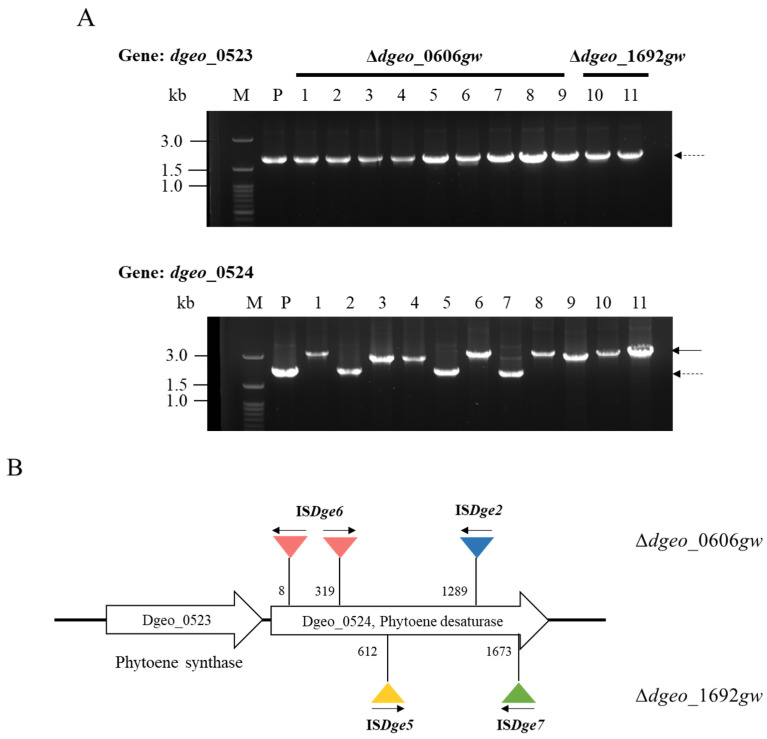
(**A**) PCR detection of transposition loci on *dgeo*_0523 and *dgeo*_0524 genes for carotenoid biosynthesis of ∆*dgeo*_0606*gw* and ∆*dgeo*_1692*gw* mutants as a result of gamma irradiation treatment. PCR was performed using primers that encompassed both target genes. Lanes: M, size marker; P, parent; 1–9, ∆*dgeo*_0606*gw* mutant; 10 and 11, ∆*dgeo*_1692*gw* mutant. Dash arrow, parent gene products; solid arrow, IS-integrated gene products with about 1 kb. (**B**) Scheme of comparative novel IS locations in which the transposition of ISs occurred in ∆*dgeo*_0606*gw* and ∆*dgeo*_1692*gw* mutants subjected to gamma irradiation. Numbers indicate the integrated loci of the genes. Arrows indicate the transcriptional direction of Tpase in the IS elements.

**Figure 4 microorganisms-12-00328-f004:**
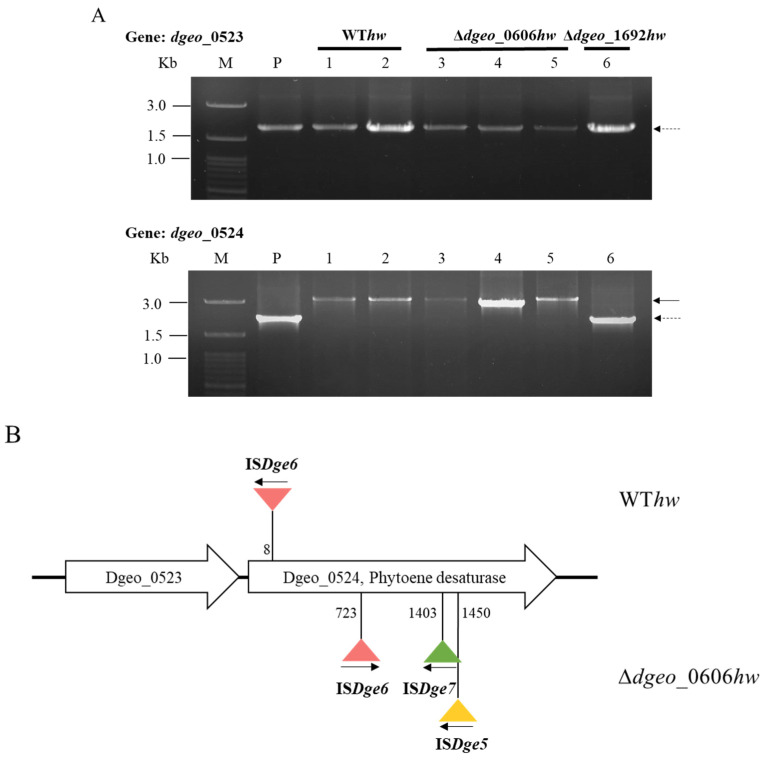
(**A**) PCR detection of transposition loci on two target genes, *dgeo*_0523 and 0524, for carotenoid biosynthesis of WT*hw* and ∆*dgeo*_0606*hw* mutant subjected to hydrogen peroxide treatment. PCR was performed using a primer involved in the carotenoid main pathway. Lanes: M, size marker; P, parent (WT); 1 and 2, WT*hw*; 3–5, *dgeo*_0606*hw* mutant; 6, ∆*dgeo*_1692*hw* mutant. Dash arrow, parent gene products; solid arrow, IS integrated gene products of about 1 kb. (**B**) Scheme of comparative novel IS locations in which the transposition of ISs occurred in non-pigmented colonies of WT*hw* and ∆*dgeo*_0606*hw* mutant subjected to hydrogen peroxide treatment. Numbers indicate the integrated loci of the genes. Arrows indicate the transcriptional direction of Tpase in the IS elements.

**Figure 5 microorganisms-12-00328-f005:**
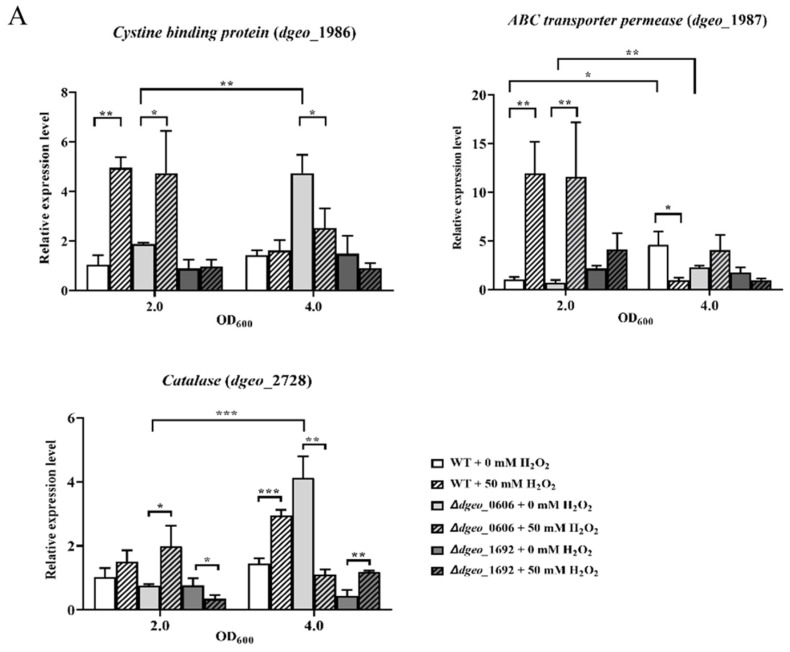
(**A**) qRT-PCR was conducted for each gene in the WT, ∆*dgeo*_0606 and ∆*dgeo*_1692 strains for the expression levels of cystine-binding protein, ATP-binding cassette (ABC) transporter permease, and catalase under absent and present H_2_O_2_ conditions of 50 mM. (**B**) qRT-PCR was conducted for each transposase family in the WT and ∆*dgeo*_0606 mutant strains for the expression levels of IS*Dge2*, IS*Dge5*, IS*Dge6*, and IS*Dge7* transposases under absent and present H_2_O_2_ conditions of 50 mM. Pair-wise comparisons between experimental groups were conducted via Student’s *t*-test with the Prism^TM^ ver. 8.0 software (* *p* < 0.05; ** *p* < 0.01; *** *p* < 0.001; **** *p* < 0.0001).

## Data Availability

Data are contained within the article and Appendix A.

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
