# Peer review of "The Transposition of Insertion Sequences in Sigma-Factor- and LysR-Deficient Mutants of Deinococcus geothermalis"

_microorganisms, 2024, doi:10.3390/microorganisms12020328_

Round 1

Reviewer 1 Report (Previous Reviewer 2)

Comments and Suggestions for Authors

The manuscript described on the IS transposition in a sigma factor and LysR deficient mutant strain under stresses related to oxidations.

 This manuscript is well written, and the experiments are appropriately designed to clarify the functions of sigma factor and LysR. However, the content of the paper does not go beyond the realm of phenomenology. This reviewer looks forward to deep considerations based on the results.

There are concerning as follows:

1.       It is difficult to find advanced points considering of previously reported studies that related to this subject.

2.       Although the authors suggested that this study was performed to understand the mechanisms and functions the sigma factor Dgeo_0606 and LysR family regulator in the “Introduction”. However, based on the results of the experiment, the core details regarding mechanisms and functions have not been clarified.   

Author Response

There are concerning as follows:

  1. It is difficult to find advanced points considering of previously reported studies that related to this subject.

> Due to the experimental limitations of this work, non-pigment phenotypic selection for active IS transposition in Deinococcus strains has been employed. This approach has been extensively applied to various target gene-disrupted mutants of D. geothermalis. The active transposition of IS elements has revealed a complex regulatory system dependent on intracellular redox imbalance and an unclear regulatory network. Therefore, it is necessary to define a systematic network regulation using different oxidation resources and target gene disruption at different growth phases. This work highlights that two target gene-disrupted mutants exhibited specific IS family member transpositions upon gamma irradiation and H2O2 treatment. The transposed IS elements overlapped with those from previous gene disruptions and oxidation resources. Accumulating current data on the regulatory network for IS transposition could potentially explain bacterial genome plasticity and evolution facilitated by IS transposition. Thus, the collection of IS transposition data presents a new challenge in navigating the complex path forward.

  1. Although the authors suggested that this study was performed to understand the mechanisms and functions the sigma factor Dgeo_0606 and LysR family regulator in the “Introduction”. However, based on the results of the experiment, the core details regarding mechanisms and functions have not been clarified.

> Indeed, this work does not primarily address the mechanisms and functional roles of a putative sigma factor, Dgeo_0606, and a LysR family transcriptional regulator. However, a portion of the experimental data suggests that phenotypic physiological differences arise due to target gene disruption. Specifically, the focus lies on whether IS transposition occurs under target gene-disrupted conditions amid two distinct oxidative stresses. Therefore, the sentences from lines 103 to 106 have been revised to center the discussion around this aspect of the research.

Reviewer 2 Report (Previous Reviewer 3)

Comments and Suggestions for Authors

This manuscript is a re-submission of the once-rejected "microorganisms-2782135" and has been well revised and improved, responding to most of review comments.

However, some review comments were not responded to. For example, regarding the new Figure 2B (previous Figure 2C), the following comment was provided:

4) ... A wild-type transformant with “the empty pRADgro”  showed increased viability against H2O2 as indicated in the leftmost insets of Figure 2C, but the opposite is seen... The authors should refer to these contrasting effects of “the empty pRADgro”.

The authors' response provided no indication of line numbers in the revised manuscript, just stating "Thanks for suggestion and correction. The sentences were improved following reviewer’s suggestion," which is not an honest manner. 

Overall evaluation of the re-submitted manuscript is "Minor revision".

Author Response

However, some review comments were not responded to. For example, regarding the new Figure 2B (previous Figure 2C), the following comment was provided:

4) ... A wild-type transformant with “the empty pRADgro” showed increased viability against H2O2 as indicated in the leftmost insets of Figure 2C, but the opposite is seen... The authors should refer to these contrasting effects of “the empty pRADgro”.

> Firstly, I apologize to the reviewer for any unclear responses to their comments. In Figure 2B, two distinct growth phases are depicted: the upper panel illustrates an OD of 2.0 for the early exponential phase without the expression vector pRADgro, while the lower panel illustrates an OD of 4.0 for the late exponential phase with the pRADgro vector present. Both growth stages are pivotal in defining the intracellular redox imbalance within growing Deinococcus cells. The early exponential phase cells exhibit a higher reduction capacity, whereas cells in the late exponential phase possess a higher oxidation state compared to their respective conditions. This redox status significantly influences IS transpositions and the cells' survival capacity under oxidative stress conditions. Additionally, intracellular levels of anti-stress redox potential substances or antagonistic factors, such as anti-sigma factors, vary in a growth phase-dependent manner. Despite WT cells demonstrate markedly different viability between the early and late exponential phases when subjected to H2O2 treatment, the viability levels exhibiting some flexibility regarding the inhibitory concentration of H2O2 in independent trials (blow data; viability test of three strains at OD4.0). The pRADgro vector did not directly affect viability against H2O2 treatment. Consequently, the sentence has been revised in lines 255-263 to reflect these considerations.

The authors' response provided no indication of line numbers in the revised manuscript, just stating "Thanks for suggestion and correction. The sentences were improved following reviewer’s suggestion," which is not an honest manner. 

>The authors presumed that the reviewer was encountering the most recent resubmitted manuscript for the first time. As a result, authors integrated the revised designations throughout the entire text and provided the revised sentence without indicating its corresponding row number. Nonetheless, in this revision, authors included the row numbers in the revised sentence.

This manuscript is a resubmission of an earlier submission. The following is a list of the peer review reports and author responses from that submission.

Round 1

Reviewer 1 Report

Comments and Suggestions for Authors

The manuscript focus on the effects of the gamma radiation and oxidative stress exposure in D. geothermalis bacteria. Specifically. its association to the transposition of IS elements by the creation of mutants strains.

I think the research is very interesting and the results/discussion sections are consistent and solid, reflecting a great amount of work. In this line, I only suggest to move some figures to the supplementary, such as figure 1, and some of figure 2.

Reviewer 2 Report

Comments and Suggestions for Authors

This manuscript described on the physiological role of a sigma factor and a LysR by using gene-deficient mutant.

1.     Although this manuscript is well written and the experiments are well designed, the conclusion of the physiological role of the sigma factor and the LysR was not described in the abstract. If possible, please include a conclusion as a summary of the entire research at the end.

2.     Please discuss the advanced points considering of the background study in the first paragraph of “Discussion”.

3.     Please discuss the limitations of the study. These will aid in discussing the scope for future studies.  

Reviewer 3 Report

Comments and Suggestions for Authors

Comments on the Quality of English Language

Some moderate editing is needed.